# Omni-Seg: A Single Dynamic Network for Multi-label Renal Pathology Image Segmentation using Partially Labeled Data

**Ruining Deng**[1]                      R.DENG@VANDERBILT.EDU

**Quan Liu**[1]                      QUAN.LIU@VANDERBILT.EDU

**Can Cui**[1]                      CAN.CUI.1@VANDERBILT.EDU

**Zuhayr Asad**[1]                    ZUHAYR.ASAD@VANDERBILT.EDU

**Haichun Yang**[2]                   HAICHUN.YANG@VUMC.ORG

**Yuankai Huo**[1]                   YUANKAI.HUO@VANDERBILT.EDU

[1] *Vanderbilt University, Department of Computer Science, Nashville, TN, USA 37215*

[2] *Vanderbilt University Medical Center, Department of Pathology, Nashville, TN, USA 37232*

**Editors:** Under Review for MIDL 2022

## Abstract

Computer-assisted quantitative analysis on Giga-pixel pathology images has provided a new avenue in histology examination. The innovations have been largely focused on cancer pathology (i.e., tumor segmentation and characterization). In non-cancer pathology, the learning algorithms can be asked to examine more comprehensive tissue types simultaneously, as a multi-label setting. The prior arts typically needed to train multiple segmentation networks in order to match the domain-specific knowledge for heterogeneous tissue types (e.g., glomerular tuft, glomerular unit, proximal tubular, distal tubular, peritubular capillaries, and arteries). In this paper, we propose a dynamic single segmentation network (Omni-Seg) that learns to segment multiple tissue types using partially labeled images (i.e., only one tissue type is labeled for each training image) for renal pathology. By learning from 150,000 patch-wise pathological images from six tissue types, the proposed Omni-Seg network achieved *superior segmentation accuracy* and *less resource consumption* when compared to the previous the multiple-network and multi-head design. In the testing stage, the proposed method obtains "completely labeled" tissue segmentation results using only "partially labeled" training images. The source code is available at https://github.com/ddrrnn123/Omni-Seg.

**Keywords:** Renal pathology, Image segmentation, Multi-label, Self-supervised Learning

## 1. Introduction

The recent advances in digital pathology, and particularly the approach of whole slide imaging (WSI), have led to a paradigm shift in pathology (Bandari et al., 2016). Computer-assisted tumor segmentation has been broadly used in cancer pathology (Nguyen et al., 2021). However, there are increasing demands in analyzing more comprehensive tissue types beyond the tumor, including investigating tumor micro-environments and non-cancer pathology (Rangan and Tesch, 2007) rather than cancer pathology. Briefly, the learning algorithms can be asked to examine more comprehensive tissue types simultaneously, as a multi-label setting. Moreover, the recent advances in data sharing are providing increasingly large amounts of publicly available data, which requires a more robust computer-assisted analytics tool for large-scale multi-center and multi-stain WSIs (Ginley et al., 2019). Ideally,

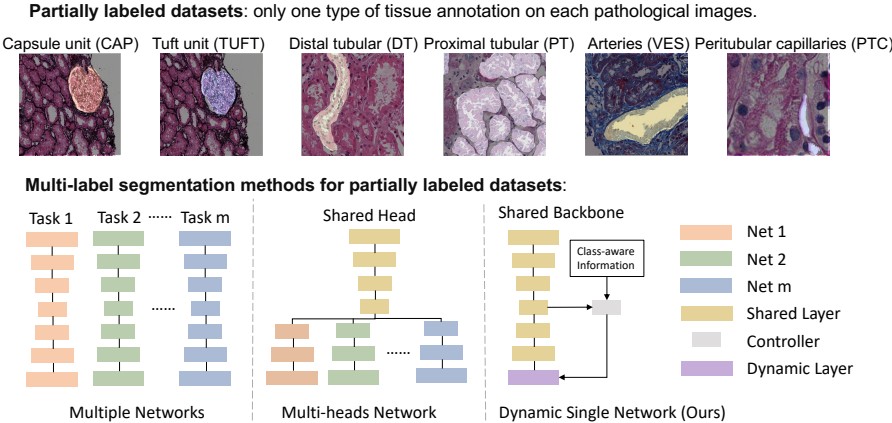

Figure 1: **Multi-label pathology dataset** – Due to being labor-intensive and time-consuming, there is only one manual annotation for one type of tissue on pathological images. A majority of prior arts in renal pathology trained multiple independent networks for different tissue types, increasing computational complexity. More recently, multi-head networks were proposed for the partial label dataset but were inflexible for a new task. In this paper, we proposed a single network with a single dynamic head using a class-aware controller and dynamic head mapping to achieve multi-label pathological segmentation efficiently.

the computer-assisted segmentation method should be able to segment all types of tissue on large-scale pathological images in order to alleviate the labor-intensive, time-consuming manual annotation (Barisoni et al., 2017; Wernick et al., 1993).

One major hurdle in achieving automated multi-label pathological image segmentation is the so-called partially labeling issue. This issue is caused by intensive labor and resource cost in pixel-level manual annotations on Giga-pixel pathological images, so that many datasets contain annotations of only one type of tissue. Mainstream approaches tackle the partial label issue by splitting the partially labeled cohorts into several fully labeled subsets and training "multiple networks" for different tasks (Yu et al., 2019; Isensee et al., 2019; Zhang et al.; Myronenko and Hatamizadeh, 2019; Zhu et al., 2019). This intuitive strategy, however, increases the computational complexity dramatically. Another more recent family of solutions is to employ multi-head networks. A multi-head network typically consists of a shared encoder, and several task-specific decoders (heads) (Chen et al., 2019; Fang and Yan, 2020; Shi et al., 2021). Moreover, the multi-class segmentation method with a single network (González et al., 2018; Cerrolaza et al., 2019) shows the promising performance for partially labeled datasets. However, the redundant implementation of heads is not only wasteful but also inflexible, especially when it is extended to a new task. To mitigate the redundancy, Zhang et al. (Zhang et al., 2021) proposed a dynamic on-demand network (DoDNet) for radiological image analysis, which is an encoder-decoder network with a dynamic head that segments multiple organs and tumors as previously done by multiple

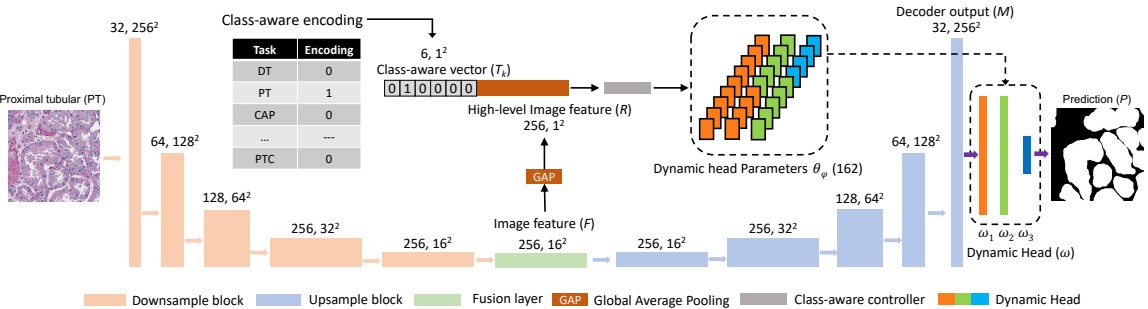

Figure 2: **Omni-Seg pipeline** – The proposed method consists of a residual U-Net backbone, a class-aware controller, and a single dynamic segmentation head. A class-aware knowledge encoder is implemented for domain-specific knowledge learning from the multi-label dataset.

networks or heads. However, these proposed methods are designed for 3D radiological data with multiple pre-trained models available. Unfortunately, it is still an unsolved problem on how to design a single multi-label segmentation model to obtain completely annotated pathological images only using partially annotated images.

In this paper, we propose a single segmentation network (Omni-Seg) which is optimal for pathology that learns to segment multiple tissue types at their optimal pyramid scales using partially labeled images (i.e., only one tissue type is labeled for each training image) in order to provide an efficient multi-label segmentation approach for pathology quantification. Briefly, our proposed method innovative contribution is three-fold: (1) To our knowledge, Omni-Seg is the first "single" dense object segmentation network for renal pathology using only a partially labeled dataset with large variations in magnification; (2) The proposed dynamic segmentation network achieves superior segmentation performances with fewer parameters compared with current multi-network approaches; (3) In the testing stage, the proposed method obtains "completely labeled" tissue segmentation results using only "partially labeled" training images. We benchmarked the existing methods on the latest publicly available tissue segmentation dataset in renal pathology to enable reproducible research, quick adaptations for other groups, and a new state-of-the-art performance as a reference for future studies. The source code has been made publicly available at https://github.com/ddrrnn123/Omni-Seg.

## 2. METHOD

The overall framework of the proposed Omni-Seg method is presented in Fig.2.

### 2.1. Dynamic multi-label modeling

In partially labeled datasets (Jayapandian et al., 2021), only one tissue type might be provided by each training image, which cannot be directly used for training a canonical multi-label segmentation network. In the proposed method, we assign the class-aware knowledge

with different tissue types to a $m$-dimensional one-hot vector for class-aware encoding (Chen et al., 2017b). The $m$ is the number of tissue types, The encoding calculation is presented in the following equation:

$$T_k = \begin{cases} 1, & if \quad k = i \\ 0, & otherwise \end{cases} \quad k = 1, 2, ..., m \tag{1}$$

where $T_k$ is a class-aware vector of $i$th tissue.

To encode domain-specific information to the embedded features, we merge the class-aware vector into the low dimensional feature embedding at the bottom of the residual U-Net architecture. The image feature $F$ is summarized by a global average pooling (GAP) via the feature vector $\mathbb{R}^{N \times 256 \times 1 \times 1}$, where $N$ is batch-size. The class-aware $T_k$ vector is reformed to be the same shape as the image features for the following fusion step. The high-level image features and the class-aware vector are concatenated. Then, a single 2D convolutional layer called the class-aware controller $\varphi$ is employed as a feature-based fusion block to refine the fusion features as the final controller for the dynamic head mapping; this follows in the next equation:

$$\omega = \varphi(GAP(F||T_k; \Theta_\varphi) \tag{2}$$

The input is the combination of $F$ and $T_k$ with the fusion process $||$, while the $\Theta_\varphi$ is the output channel of the class-aware controller, which also defines the parameter size for the dynamic head. Therefore, both the computational and spatial complexity of our task encoding strategy is lower than that seen in the multi-network strategy.

## 2.2. Dynamic head mapping

Inspired by DoDNet (Zhang et al., 2021), a binary segmentation network is employed to achieve multi-label segmentation via a dynamic filter. From the above multi-label modeling, we achieve joint low-dimensional image feature vectors and class-aware vectors at optimal segmentation magnification. Then, such information is mapped to control a light-weight dynamic head in order to specify the targeting tissue type from the inputs.

The dynamic head concludes with three layers. The first two have eight channels, while the last layer has two channels. The amount of the parameters for the dynamic head is 162. We directly map the parameters from the fusion-based feature controller to the kernels in the dynamic head to achieve precise segmentation by multi-modality features. As such, the filtering process can be defined in Eq.3

$$P = ((((M * \omega_1) * \omega_2) * \omega_3) \tag{3}$$

where $*$ is convolution, $M$ is the output from the decoder, $P \in \mathbb{R}^{N \times 2 \times W \times H}$ is the final prediction, while $N$, $W$, and $H$ are the batch-size, width, and height of the dataset.

## 2.3. Residual U-Net backbone

For the implementation of the segmentation backbone, we utilize the residual U-Net as the segmentation unit according to DoDNet(Zhang et al., 2021), as shown in Fig.2. Different from the 3D network design in DoDNet, we develop the 2D convolutional blocks with

kernel size 3×3 to be compatible with 2D pathological images. All the convolutional blocks are followed with ReLU activation after a group normalization (Wu and He, 2018). A convolutional fusion layer with kernel size 3×3 is employed to learn high-level image features. Each downsampling block is implemented with a stride of 2 to halve the scale of input feature maps. Therefore, the feature maps are adjusted by multiple encoder-decoder blocks in pyramid scales. Symmetrically, the decoder upsamples the feature maps with an upsample factor of 2 and halves the channel number. In each upsample block, a low-level feature map (as a skip connection from the corresponding encoder layer) is summed with an upsampled feature map and then refined by a residual block. Following this feature-based learning, the output concludes high-level features for segmentation.

### 2.4. Testing stage aggregation with complete labels

In the testing stage, large-scale pathological images in 40 × magnification are captured and split into 256 × 256 patches at the optimal scales (Jayapandian et al., 2021) with the downsampling process for different tissues. The same scales and methods (downsample and randomly crop) as those in (Jayapandian et al., 2021) were kept to ensure a consistent comparison. The class-aware controller is switched to receive six partially labeled segmentation for different tissue structures on patch-wise images. All of the segmentation is aggregated and mapping on the 40 × pathological images as a completely labeled tissue segmentation for six tissue types.

## 3. Data and Experiments

### 3.1. Data

1751 regions of interest (ROIs) images were captured from 459 WSIs, obtained from 125 patients with minimal change diseases. The images were manually segmented for six structurally normal pathological primitives (Jayapandian et al., 2021), using the digital renal biopsies from a multi-center Nephrotic Syndrome Study Network (NEPTUNE) (Barisoni et al., 2013). All of the images were 3000×3000 pixels in 40× magnification, including glomerular tuft (TUFT); glomerular unit (CAP); proximal tubular (PT); distal tubular (DT); peritubular capillaries (PTC); and arteries (VES) in Hematoxylin and Eosin (H&E), Periodic-acid-Schiff (PAS), Silver(SIL), and Trichrome (TRI) stain. Four stain methods were regarded as color augmentations in each type of tissue. We followed (Jayapandian et al., 2021) to randomly crop all the images and downsample them into 256×256 patches for the optimal magnifications. To resolve the unbalanced issue, we calculated the amount of each type of tissue and normalized the cropped patch numbers from the original images. Using the publicly available dataset, we kept the same splits as its original release in (Jayapandian et al., 2021) that randomly split the entire dataset into training, validation, and testing sets, following the ratio of 6:1:3. The splits were performed at the patient level to avoid data contamination. One sample randomly selected in-house samples from human nephrectomy tissues with 3 $\mu$m thickness sections cut and stained with Periodic acid–Schiff (PAS) were used for completely labeled segmentation aggregation in the testing stage.

Table 1: Performance of different methods on the multi-label dataset. Dice similarity coefficient (%, the higher, the better), Hausdorff Distance (Micron unit, the lower, the better), and Mean Surface Distance (Micron unit, the lower, the better) are used for evaluation. The red mark indicates the best performance.

| Method | | DT | | | PT | | | CAP | | |
|---|---|---|---|---|---|---|---|---|---|---|
| | | Dice↑ | HD↓ | MSD↓ | Dice↑ | HD↓ | MSD↓ | Dice↑ | HD↓ | MSD↓ |
| Multi U-Net (Jayapandian et al., 2021) | | 78.51 | 107.63 | 36.05 | 88.25 | 63.84 | 8.53 | 95.42 | 54.38 | 9.34 |
| Multi DeepLabV3 (Chen et al., 2017a) | | 77.92 | 107.61 | 35.45 | 88.49 | 60.24 | 9.05 | 95.78 | 50.42 | 9.76 |
| TAL (Fang and Yan, 2020) | | 47.76 | 280.44 | 198.07 | 48.49 | 179.41 | 84.91 | 51.82 | 402.76 | 272.76 |
| Med3D (Chen et al., 2019) | | 47.73 | 194.55 | 110.09 | 35.80 | 217.41 | 109.51 | 89.49 | 89.76 | 19.92 |
| Multi-class (González et al., 2018) | | 47.76 | 280.44 | 198.07 | 88.36 | 64.84 | 8.85 | 95.93 | 85.64 | 11.1 |
| Omni-Seg(Ours) | | 81.01 | 97.27 | 24.19 | 89.80 | 57.11 | 6.86 | 96.50 | 53.08 | 7.30 |

| Method | | TUFT | | | VES | | | PTC | | | Average | | |
|---|---|---|---|---|---|---|---|---|---|---|---|---|---|---|
| | Dice↑ | HD↓ | MSD↓ | Dice↑ | HD↓ | MSD↓ | Dice↑ | HD↓ | MSD↓ | Dice↑ | HD↓ | MSD↓ |
| Multi U-Net (Jayapandian et al., 2021) | 96.05 | 63.16 | 9.72 | 77.66 | 101.59 | 54.30 | 72.73 | 31.80 | 13.53 | 84.77 | 70.4 | 21.91 |
| Multi DeepLabV3 (Chen et al., 2017a) | 96.45 | 51.34 | 7.16 | 81.08 | 84.31 | 44.69 | 72.69 | 30.75 | 14.27 | 85.40 | 64.11 | 20.06 |
| TAL (Fang and Yan, 2020) | 76.95 | 137.18 | 65.58 | 47.67 | 244.11 | 191.25 | 49.37 | 52.15 | 35.79 | 53.67 | 216.01 | 141.39 |
| Med3D (Chen et al., 2019) | 92.80 | 80.84 | 14.80 | 58.46 | 150.71 | 78.77 | 49.78 | 44.53 | 27.24 | 62.34 | 129.63 | 60.06 |
| Multi-class (González et al., 2018) | 46.63 | 486.30 | 359.04 | 47.67 | 244.12 | 191.20 | 49.28 | 64.38 | 49.41 | 62.69 | 204.28 | 136.27 |
| Omni-Seg(Ours) | 96.59 | 44.36 | 6.20 | 85.05 | 71.13 | 21.99 | 77.23 | 26.49 | 8.15 | 87.70 | 58.24 | 12.45 |

## 3.2. Experimental details

Six image pools were designed for different tissues to organize training batches with the same tissue type patches by following the image pool from Cycle-GAN (Zhu et al., 2017). The batch size was four, while the image pool size was eight. Once one pool stacked the number of images to more than the batch size, the images in the pool were queried out from the pool and fed into the network. Binary Dice loss and cross-entropy loss were combined during each backpropagation for all the tasks as the loss function. The boundaries of ground truth were assigned a higher weight with 1.2 when calculating the loss to emphasize the boundary prediction accuracy, which is widely implemented in segmentation (Ronneberger et al., 2015). Stochastic Gradient Descent (SGD) was used for weight update with a learning rate of 0.001 and 0.99 decay. The general data augmentation including Affine, Flip, Contrast, Brightness adjustment, Coarse Dropout, Gaussian Blur, and Gaussian Noise in the imagaug package (Jung et al., 2020) was implemented for the whole training dataset in all of the methods, with a probability of 0.5.

The Dice similarity coefficient (Dice), Hausdorff distance (HD), and Mean Surface Distance (MSD) were used as performance metrics for this study. The distance metrics are in the Micron unit. For $40\times$ images, each pixel is equal to 0.25 Micron. The mean of the Dice similarity coefficient from six tasks was used to select the optimal model in the validation dataset, and the performances were evaluated in the testing dataset. The performances selected the best models in testing within 100 epochs. All the experiments were completed on the same workstation, with the NVIDIA Quadro P5000 GPU.

## 4. Results

We compared the proposed Omni-Seg with baseline models, including (1) multiple individual U-Net models (Multi U-Net) (Jayapandian et al., 2021), (2) multiple individual

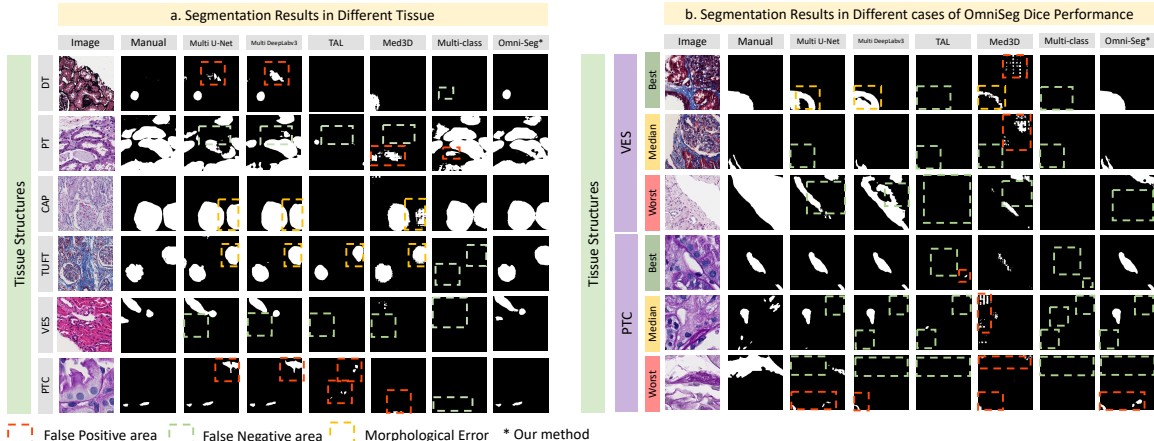

Figure 3: **Qualitative results** – Figure 3 displays the qualitative results from different baseline methods.The red bounding box shows the false positive prediction. The green bounding box shows the false negative prediction. The yellow box indicates the morphological error in the prediction.

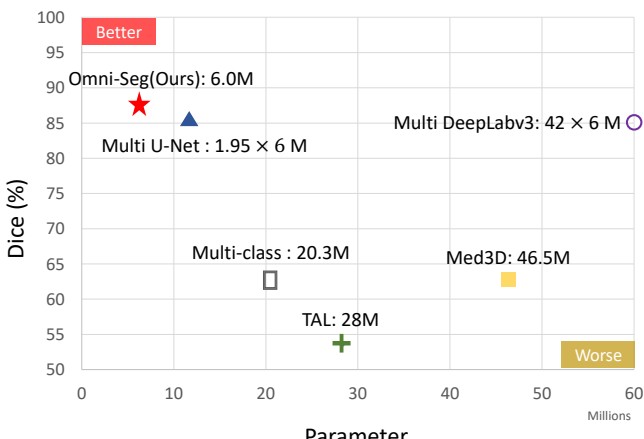

Figure 4: **Parameter usage** – This figure shows the relationship between Dice and parameter usage for each method. Using fewer parameter to achieve a higher Dice score is better.

DeepLabv3 models (Multi DeepLabv3) (Chen et al., 2017a) trained on each partially labeled dataset to solve the multi-label issue independently, (3) One multi-head model with target adaptive loss (TAL) for multi-class segmentation (Fang and Yan, 2020), (4) one multi-head 3D model (Med3D) for multiple partially labeled datasets (Chen et al., 2019), one multi-class segmentation model for partially labeled datasets (González et al., 2018).

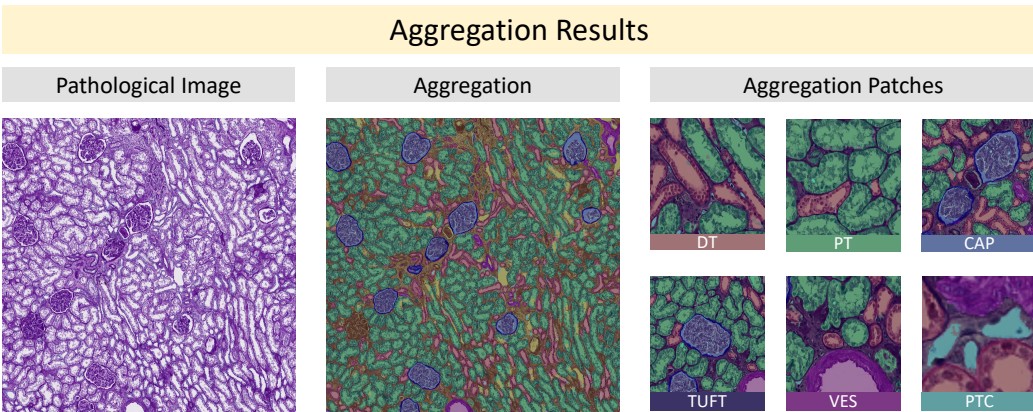

Figure 5: **Multiple segmentation aggregation** – This figure shows the "completely labeled" segmentation of six tissue structures from "partially labeled" training images. The yellow areas are other tissue types.

Table 1 shows the performance metrics for the segmentation of each tissue over the whole dataset. Figure 3.(a) shows the performance of different methods on the multi-label dataset. Figure 3.(b) demonstrates the cases of the best, the median, and the worst dice score in OmniSeg in two tissue types with consistent results from different methods. Figure 4 displays the relationship between parameter number and Dice score in each method.

As a result, training the whole dataset on a single multi-label network aggregated more domain-specific knowledge on pathological images, achieving better results than individual networks (i.e., Multi U-Net, Multi DeepLabv3). Due to missing the perspectives from other tissues, the results from individual networks had more false-positive predictions (the red bounding box) for the similar structures among different tissues and more false-negative predictions (the green box) from the incomplete dataset.

Our single dynamic network also led to "completely labeled" tissue segmentation results from only "partially labeled" training images on each histological image in Figure 5. The overlap areas of tufts and capsules will be displayed as tufts. For the remaining multiple foreground pixels, we labeled them as background. Only qualitative results are presented since the manual annotation is not available. The large-scale multi-label segmentation aggregation generates quantitative estimation for clinical examination.

## 5. Conclusion

In this paper, we propose a holistic segmentation network (Omni-Seg) that segments multiple tissue types using partially labeled images via a single dynamic head. The proposed method achieves superior segmentation performance with less computational resource consumption. In the testing stage, our single dynamic network achieved "completely labeled" tissue segmentation aggregation from only "partially labeled" training images.

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
