# OpenReview forum: "Omni-Seg: A Single Dynamic Network for Multi-label Renal Pathology Image Segmentation using Partially Labeled Data"
_MIDL.io/2022/Conference — MIDL 2022_

### Official Review · Reviewer_qNzf · 2022-01-23

**Confidence:** 5
**Preliminary Rating:** 4
**Recommendation:** Poster

**Summary:**

The authors present a multi-class segmentation network trained with partially labeled datasets in the context of renal pathology image segmentation. Instead of training multiple networks or a multi-head network, the authors modulate a set of final convolution layers (so-called dynamic layer) according to the class that is required for the network to segment. The parameters of such layer are modulated by the image features of the lowest (most reduced) layer of a unet segmentation architecture, to which a one-hot encoding of the class to be segmented is concatenated.

**Strengths:**

Great evaluation. Interesting method. The problem of having a multiple class segmentation method from partially labeled datasets is of value to the community. No current solution is satisfactory. Of potential value to the community.

**Weaknesses:**

Literature review. The authors seem to ignore the multi-organ segmentation literature, where training with partial labels is performed. See for instance Gonzalez18 for a multi-class segmentation method with a single network with partial labelled datasets or a review of the methods Cerrolaza19. The paper would benefit from the inclusion of such methods on the review.
Clarity of the method.
-	It is unclear what the function phi of Eq. 2 is. Also, Theta sub-Phi is not clearly defined.
-	While the authors state that there is only allowed one label to each image path, it would be interesting to discuss how the inclusion of a plurality of labels to each of them would be treated in this method.
-	In Equation 3, the authors state that the dynamic head is just a set of convolutions. It seems that there is no activation function post each of the convolutional layers. If that is the case, the three convolutional layers could be simplified into a single convolution, since everything is linear. Please clarify.
-	Section 2.4. There are several references to “optimal scales”. I assume that the images are captured at 40x magnification and split into 256x256 patches. The downsample method “for different tissues” is not clear.
-	Data management. The authors claim that “We randomly split the entire dataset into training, validation and testing sets”. This random split is worrisome. Nearby patches will share similar image characteristics and therefore can be easy to classify. The split should always be done in an WSI-based randomization (not patch-based). Please clarify if the random split is done in a patch-based or WSI based method. Even more, it should be done on a patient-based manner.
-	The authors state that “Binary Dice loss and cross-entropy loss were used as the loss function”. Is it a combination of both? Does it depend on the experiment? Please clarify.
-	When generating “completely labeled” tissue segmentations, the authors will pass each patch multiple times through the network with different class encodings on their class-aware vector. How do the authors deal with the case that a pixel obtains multiple foreground hits? There is no information on the paper.
Performance of the method in comparison to others. The authors compare their method to multiple networks or multi-heads networks, but do not compare to multiple class segmentation networks (see Gonzalez 18). The authors’ method need a pass of the network for each class  at test time, while multiple-class segmentation methods require a single pass to all classes.

Gonzalez18. González, Germán, George R. Washko, and Raúl San José Estépar. "Multi-structure segmentation from partially labeled datasets. application to body composition measurements on ct scans." Image Analysis for Moving Organ, Breast, and Thoracic Images. Springer, Cham, 2018. 215-224.
Cerrolaza19. Cerrolaza, Juan J., et al. "Computational anatomy for multi-organ analysis in medical imaging: A review." Medical image analysis 56 (2019): 44-67.


**Deanonymize Review:**

no

**Detailed Comments:**

The use of the term precision medicine in the abstract is not clearly motivated.
Page 2. “Should be able to segmentation” – “should be able to segment”
In Page 4 stating that “The image feature F is refined” seems incorrect, “summarized” is probably a better verb to define what is done.
Page 6. “in clinical research” seems not to be needed.
Fig.3 There is no yellow arrow. There are boxes.
It would be of interest to see a patch with complete annotations and the different outputs of the network in detail. One zoom on a patch of Figure 5 shown in the manner of Figure 3.


**Paper Type:**

validation/application paper

**Questions To Address In The Rebuttal:**

I would like the authors to address all the weaknesses described above. This will increase the clarity and value of the paper. Special interest is the data handling. If the randomization is patch-based, then the results are heavily biased and the paper can not be published in its form.

**Special Issue:**

no

---

### Official Review · Reviewer_EjGQ · 2022-01-24

**Confidence:** 4
**Preliminary Rating:** 4
**Recommendation:** Poster

**Summary:**

This paper presents a multi-class segmentation method that uses a single network that can be trained using partially labelled datasets, a known challenge for semantic segmentation methods. The method is applied to whole-slide histopathology images. The key idea is to use a class-aware controller, which consists of a one-hot encoded vector, and which is merged into the low dimensional feature embedding at the bottom of the residual U-Net architecture. This information is then subsequently mapped to control a "dynamic" head in order to segment different tissue types from the inputs, which are then aggregated to produce a final output image. The paper presents experiments on a dataset of around 1700 ROIs obtained from 459 WSIs, and shows to outperform multiple U-Nets or multiple individual DeepLabv3 models.

**Strengths:**

- Well written paper with a clear rationale, open-source code, and nice illustrative images.
- Important problem in the field, as partially labelled data is abundantly available.
- The approach is benchmarked against important baseline approaches, U-Net and DeepLabv3.

**Weaknesses:**

- The method has a lot of overlap with the DoD-Net paper so I feel the technical novelty of this paper is limited. Also, the authors do not clearly state the technical novelty of the paper in comparison to the DoD-net paper. However, the application to WSI images is novel AFAIK.
- The experiments are conducted on a dataset which is not publicly available. It would be great to benchmark this approach using public datasets so that future approaches can also be tested against the same benchmark. Is it possible to publicly release (part of) the data and set up a challenge around this?
- Why is this approach not compared to a multi-head design? That would be a great comparison and is lacking now. This is a pity because the introduction introduces the multi-head approach as an alternative strategy so nicely.


**Deanonymize Review:**

no

**Detailed Comments:**

- Figure 3 seems to be cherrypicked because the differences between the approaches are larger than Table 1 suggests. Please add how the images are selected. Consider to select a consecutive set of 5 images with the best OmniSeg performance and the worst OmniSeg performance.

**Final Rating After The Rebuttal:**

4: Weak Accept

**Justification Of The Final Rating:**

The authors have extended the paper and addressed my concerns well, and I want to thank the authors for that. I still think that the method has substantial overlap with the DoD-Net paper so I feel the technical novelty of this paper is limited, so I am sticking to my original rating of this paper.


**Paper Type:**

methodological development

**Questions To Address In The Rebuttal:**

- Why no experiments with multi-head approaches?
- Please clearly specify the contributions of this paper in comparison to the DoDNet paper.
- Would the authors be open to organize a challenge around this dataset and this topic?

**Special Issue:**

no

---

### Official Review · Reviewer_RFRK · 2022-01-26

**Confidence:** 5
**Preliminary Rating:** 1
**Recommendation:** Poster

**Summary:**

The authors proposed a semantic segmentation approach for learns to segment multiple tissue types using partially labeled
images in renal histopathology images. The approach is based on previously proposed approach, DODNet, which consists of a single network with a single dynamic head using a class-aware controller and dynamic head mapping. The evaluation is performed on a set of regions of interest from 459 WSIs and labeled for six structurally normal pathological primitives.

**Strengths:**

- The approach addressed an important challenge in semantic segmentation in medical images: partially labeled images. Specially important in this field (histopathology)

-The method is compared with a set of currently published methods in the field.

**Weaknesses:**

- The methodological contribution is minimal. The method is practically the previously published DODNet, just input is 2D instead of 3D as initially proposed.
- The overall performance is not so convincing as stated by the authors. It seems the improvement is marginal compared to multi U-Net. It would have been also good to compare the method with a multi-head approach.

**Deanonymize Review:**

no

**Final Rating After The Rebuttal:**

3: Borderline

**Justification Of The Final Rating:**

I appreciate the authors' effort including a comparison with a multi-head approach. However, I still consider that the method has too much overlap with the DoDNet approach. Then, this paper can be considered for its contribution in the application but a more exhaustive evaluation should be performed comparing it with more current approaches and state-of-the-art solution. I have changed my rating to reflect the effort carried out by the authors.

**Paper Type:**

validation/application paper

**Questions To Address In The Rebuttal:**

- Discuss clearly the methodological contribution of the paper. Maybe this would be better as a validation paper but the validation needs to be extended: The method is practically the previously published DODNet, just input is 2D instead of 3D as initially proposed.
- Compare with multi-head approach: The overall performance is not so convincing as stated by the authors. It seems the improvement is marginal compared to multi U-Net. It would have been also good to compare the method with a multi-head approach.

**Special Issue:**

no

---

### Meta-Review · Area_Chair_ersC · 2022-02-19

**Recommendation:** Accept (Poster)
**Confidence:** 4

**Metareview:**

While two our of three reviewers pointed out that the method is very similar to a previously published approach, DODNet,  these reviewers still see value in the extensive evaluation presented in this paper, and both suggested weak accept. The first reviewer also increased the score to borderline after the rebuttal owing to additional experiments for evaluation. I also agree with the reviewers that this paper addresses an important problem in the field, i.e.  partially labelled data, and presents extensive evaluation and benchmarking against important baseline approaches, and therefore suggest acceptance of this paper.

---

### Decision · Program_Chairs · 2022-02-28

Accept